# Cuttlefish Bone-Derived Biphasic Calcium Phosphate Scaffolds Coated with Sol-Gel Derived Bioactive Glass

**DOI:** 10.3390/ma12172711

**Published:** 2019-08-24

**Authors:** Ana S. Neto, Daniela Brazete, José M.F. Ferreira

**Affiliations:** Department of Materials and Ceramic Engineering, CICECO, University of Aveiro, 3810-193 Aveiro, Portugal

**Keywords:** cuttlefish bone, biphasic calcium phosphate, porous scaffolds, sol-gel coatings, bioactivity, tissue engineering

## Abstract

The combination of calcium phosphates with bioactive glasses (BG) has received an increased interest in the field of bone tissue engineering. In the present work, biphasic calcium phosphates (BCP) obtained by hydrothermal transformation of cuttlefish bone (CB) were coated with a Sr-, Mg- and Zn-doped sol-gel derived BG. The scaffolds were characterized by X-ray diffraction, Fourier transform infrared spectroscopy and scanning electron microscopy. The initial CB structure was maintained after hydrothermal transformation (HT) and the scaffold functionalization did not jeopardize the internal structure. The results of the in-vitro bioactivity after immersing the BG coated scaffolds in simulated body fluid (SBF) for 15 days showed the formation of apatite on the surface of the scaffolds. Overall, the functionalized CB derived BCP scaffolds revealed promising properties, but further assessment of the in-vitro biological properties is needed before being considered for their use in bone tissue engineering applications.

## 1. Introduction

Bone is one of the most transplanted tissues. Autologous bone grafts are still considered the gold standard procedure, but unfortunately, their availability is limited and they are intimately associated with donor site morbidity [1]. Thus, bone tissue engineering represents an important challenge to overcome the shortcomings of bone grafts, where porous scaffolds can be combined with cells and bioactive growth factors, providing a suitable environment for tissue development [2].

Bioceramic-based scaffolds have been successfully used for bone repair and replacement. Among the different bioceramics, calcium phosphate materials (CaP) like hydroxyapatite (HA) and β-tricalcium phosphate (β-TCP) are the most commonly used ones as they are chemically similar to the inorganic part of the bone. HA is stable under physiological conditions and has a slow resorption rate and β-TCP has a more soluble phase with a lower mechanical stability. The mixtures of HA and β-TCP phases create biphasic calcium phosphate (BCP) materials and enable the control of the bioactivity and the balance between resorption and the solubilisation rates [3]. Bioactive glasses (BGs) have been used as an alternative to CaP materials. They have the capacity to strongly bond to the living tissues, through the development of a bone-like HA layer on the surface [4,5]. Moreover, BGs release Si, P or Ca ions that act as chemical stimuli for the activation of osteoprogenitor cells and, consequently, enhance bone formation. The release of these ions is also known to stimulate neovascularisation and angiogenesis and, thereby, promote bone healing [6,7]. BGs were firstly synthesized by melt-quenching technique. Nevertheless, the sol-gel approach has been gaining more relevance in the last decades [8,9,10]. When compared to melt-quenching, the sol-gel derived BGs are prepared at significantly lower temperatures and they have a higher specific surface area, which, consequently, enhances their bioactive response [11]. Apart from this, the sol-gel derived BGs can be prepared with amounts of SiO_2_ up to 90 wt.%, whilst SiO_2_ contents higher than 60 wt. % are hardly achievable by melt-quenching [12]. The high number of Si–OH groups in sol-gel derived BG is beneficial for further functionalization [13]. Moreover, the composition of sol-gel derived BG can be more easily tailored to achieve the desired physicochemical and biological properties. In this regard, the incorporation of metal ions in the silicate network have been widely investigated [14,15,16,17,18]. The incorporation of strontium (Sr) into the BGs enhances bone formation as it promotes osteoblast differentiation and simultaneously inhibits osteoclast differentiation. In addition, in-vivo studies demonstrated that the Sr–doped BGs strongly bond to the bone through the apatite layer [19,20]. The incorporation of magnesium (Mg) into BGs improves the dissolution behaviour as it promotes the disruption of the silica network [21]. Furthermore, Mg–doped BGs promote osteoblast proliferation and differentiation [22]. On the other hand, the introduction of zinc (Zn) into BGs favours osteoblast proliferation and differentiation [16]. Additionally, Zn demonstrated to exert anti-inflammatory effects [23].

The combination of CaP with bioactive glasses (BG) has received an increased interest in the bone tissue engineering field, since it develops a synergetic effect improving the properties of the bone graft materials [24,25]. It was demonstrated that the incorporation of BG into a CaP material enhances the mechanical strength through the densification of the material. In addition, the bioactivity of CaP is lower when compared to those of BGs. Thereby, a combination of these types of materials creates bone grafts with enhanced biological responses [26].

The success of a bone tissue engineering approach is highly dependent on the performance of the scaffolds. Marine skeletons, like coral and cuttlefish bone (CB), have been proposed as bone graft materials as they are mainly calcium carbonate and, thereby, can be hydrothermally converted into CaP materials [27]. The internal CB lamellae matrix has an ideal microstructure with pore sizes and interconnectivity that favors bone development [28,29]. This work aims at developing BG-coated BCP scaffolds, taking advantages of their intrinsic porous structure and enhanced bioactivity. BCP scaffolds obtained through the hydrothermal transformation (HT) of CB were coated with a Sr-, Mg- and Zn-doped sol-gel derived BG (60% SiO_2_–34% CaO–2% SrO–2% MgO–2% ZnO–2% P_2_O_5_ (mol %), and the bioactivities of the uncoated and BG-coated scaffolds were tested in-vitro in simulated body fluid. The selected BG composition attempts to obtain homogeneous and amorphous materials with a good balance between degradability and bioactivity, while exploring the various biological benefits derived from the incorporation of doping ions in the coating layer.

## 2. Materials and Methods

### 2.1. Preparation of BCP Scaffolds

The BCP scaffolds were obtained through hydrothermal transformation (HT) of the hard aragonite porous structure of cuttlefish (Sepia officinalis). Firstly, the exact amount of CaCO_3_ present in the CB sample was determined by differential thermal and gravimetric thermal analyses (DTA/TG, Labsys Setaram TG-DTA/DSC, Caluire, France, heating rate 10 °C min^−1^). According to the obtained result, the CB was cut into small pieces and it was added to the required volume of the phosphorous precursor solution, di-ammonium hydrogen phosphate [(NH_4_)_2_HPO_4_, Panreac AppliChem, Barcelona, Spain]. Subsequently, the as-prepared mixtures were sealed in poly(tetrafluorethylene) lined stainless steel autoclave. The reaction took place at 200 °C for 24 h in the oven. The as-obtained scaffolds were subjected to a heat treatment at 700 °C for 1 h and using a heating rate of 0.5 °C min^−1^ to burn the organic material followed by a sintering at 1200 °C using a heating rate of 2 °C min^−1^ and a dwelling time of 2 h.

### 2.2. Preparation of Sol-Gel Derived BG and Coating of the BCP Scaffolds

In this study, a sol-gel derived BG with the composition of 60% SiO_2_–34% CaO–2% SrO–2% MgO–2% ZnO–2% P_2_O_5_ (mol %) was used. To prepare the BG, tetra-ethyl-ortho-silicate (TEOS, C_8_H_20_O_4_Si, Sigma–Aldrich, Darmstadt, Germany) and triethyl phosphate (TEP, C_6_H_15_O_4_P, Sigma–Aldrich, Darmstadt, Germany) were mixed in 10 mL of water. The pH was adjusted to values between 1 and 2, using nitric acid (HNO_3_, Panreac AppliChem, Barcelona, Spain) and mixed for 1 h. Calcium nitrate tetrahydrate [Ca(NO_3_)_2_⋅4H_2_O, Panreac AppliChem, Barcelona, Spain] was used as the precursor of calcium. Furthermore, strontium nitrate, Sr(NO_3_)_2_, Magnesium nitrate hexahydrate, Mg(NO_3_)_2_·6H_2_O, and zinc nitrate hexahydrate, Zn(NO_3_)_2_·6H_2_O, all from Sigma-Aldrich, Germany, were used as the precursors of strontium, magnesium and zinc. These nitrate salts were dissolved in 10 mL of water and, subsequently, added to the first solution and kept under magnetic agitation for 1 h to obtain a homogenous solution. The as-obtained solution was used for coating the BCP scaffolds. For this purpose, the scaffolds were immersed in the BG solution and placed in the vacuum chamber under a pressure of 0.4 bar for 20 min. Subsequently, the impregnated scaffolds were placed in an oven at 100 °C for 24 h. The scaffolds were sintered at 700 °C for 2 h and with a heating rate of 0.3 °C min^−1^.

### 2.3. Sample Characterization

#### 2.3.1. X-Ray Diffraction

X-ray diffraction (XRD) was used to identify the crystalline phases of the BCP scaffolds derived from CB as well as to confirm the glass state of the synthesized sol-gel derived BG. The data were collected in 2θ range between 10° and 100° with a 2θ–step size of 0.0260° per second in a high resolution X-ray diffractometer (PANalytical X’Pert PRO, Almelo, Netherlands) with Cu Kα radiation (λ = 1.5406 Å) produced at 40 mA and 45 kV. The relative percentages of the crystalline phases were estimated by using the HighScorePlus, a spectra-fitting software.

#### 2.3.2. Fourier Transform Infrared Spectroscopy

The sol-gel derived BG, as well as the uncoated and coated scaffolds, were examined by the attenuated total reflection Fourier transform infrared spectroscopy (FTIR-ATR). The data were collected over the spectral range of 400–4000 cm^−1^ with a total of 256 scans and a spectral resolution of 4 cm^−1^ in a FTIR Bruker Tensor 27 equipped with a Golden Gate Single Reflection Diamond ATR.

#### 2.3.3. Dilatometry

The sintered uncoated scaffolds and the dried BG were grinded and powdered. From the obtained powder, cylindrical rods were prepared with a diameter of 4.50 mm and a length of approximately 10 mm and were used for dilatometry measurements (Bahr Termo Analyse DIL 801 L, Hüllhorst, Germany; heating rate of 5 °C min^−1^). The thermal expansion coefficient was obtained according to Equation (1),
(1)$$\alpha =\frac{1}{L_{0}}\left(\frac{\Delta L}{\Delta T}\right)$$

Here, *α* is the thermal expansion coefficient, *L*_0_ is the initial sample length, Δ*L* is the change in length and Δ*T* is the change in temperature.

#### 2.3.4. Microstructure

Micro computed tomography (µ-CT, Bruker, Billerica, MA, USA) was used to observe the interconnectivity of the raw CB microstructure. It used an exposure time of 800 ms, energy of 50 kV and intensity of 200 µA. The NRecon software was used for the reconstruction of cross-section images. The microstructure of the samples was also observed by scanning electron microscopy (SEM, Hitachi SU-70, Hitachi High-Technologies Europe, GmbH, Krefeld, Germany) using an acceleration voltage of 15 kV. The samples were coated with carbon. The energy dispersive spectroscopy (EDS) was used to examine the presence of the doping elements in the samples.

#### 2.3.5. Mechanical Properties

The uncoated and coated cubic shape scaffolds with approximately 3 mm sides were subjected to a uniaxial compressive load by using a universal testing machine (AG-IS10kN, Shimadzu, Kyoto, Japan). The samples were subjected to a maximum load of 200 N with a constant crosshead speed of 0.5 mm min^−1^ in ambient conditions. The compressive load was applied perpendicularly to the lamella of BCP scaffolds derived from CB, using 5 samples in each testing condition.

### 2.4. In-Vitro Bioactivity Test

The uncoated and coated scaffolds were immersed in simulated body fluid (SBF) solution with an ionic concentration similar to those of human blood plasma to evaluate the in-vitro scaffold’s bioactivity. For this purpose, for a period of 14 days, the samples were immersed in an orbital shaker at 37 °C. The SBF solution was prepared according to the international standard ISO 23317 but using a unified standard area of 0.5 cm^2^ mL^−1^ proposed elsewhere [30]. After the immersion period, the samples were removed from the SBF solution, rinsed with distilled water and then left to dry in a silica desiccator. Once dried, the morphology of the scaffolds was analysed by SEM to observe the bioactive layer deposited onto their surfaces.

### 2.5. Statistical Analysis

The experiments were performed in triplicated and the values have been expressed as the mean ± standard deviation (SD). The statistical analysis was performed using a Student’s t-test. P < 0.05 was considered statistically significant.

## 3. Results

### 3.1. Chemical and Structural Characterization

Figure 1 displays the XRD patterns of the sol-gel derived BG, as well as the ones regarding the scaffolds before and after coating with BG by immersion in the solution, followed by heat treatment at 700 °C for 2 h. The sol-gel derived BG exhibited essentially an amorphous character. On the other hand, the scaffolds before and after the impregnation with BG, followed by heat treatment at 700 °C for 2 h, exhibited similar patterns, with HA (# 04-015-7245) and β-TCP (# 04-006-9376) being the identified crystalline phases. As estimated by HighScorePlus software, the relative percentages of the crystalline phases presented in the scaffolds were 63.9% HA and 36.1% β-TCP before the BG coating and 64.6% HA and 35.4% β-TCP after the BG coating.

The FTIR spectra of the sol-gel derived BG and the BCP scaffolds before and after the coating with BG are shown in Figure 2. The FTIR spectra of sol-gel derived BG exhibit a characteristic broad band for the asymmetric stretching for Si–O–Si from 900 cm^−1^ and 1300 cm^−1^ and centered at 1080 cm^−1^. The peak at 800 cm^−1^ is also assigned to the stretching mode of Si–O–Si [31]. Moreover, the FTIR spectra of the uncoated and coated scaffolds are identical and exhibit the characteristic vibrational modes of –PO_4_ and –OH groups. The band present at 470 cm^−1^ denotes the ν2 mode of O–P–O bending, while the ν4 mode of O–P–O bending is present at 550 cm^−1^ and 600 cm^−1^. The band at 960 cm^−1^ is associated with ν1 P–O stretching mode. The bands located at 1000 and 1088 cm^−1^ are due to the ν3 mode. The bands at 940 and 960 cm^−1^ denote the ν1 non-degenerate P–O symmetry stretching mode. The vibrational mode of –OH group appear at 630 cm^−1^ with low intensity and the stretching mode should appear at 3575 cm^−1^ [32].

### 3.2. Microstructure

The unique layered and interconnected porosity of CB was observed by µ-CT (Figure 3a). The porous architecture of scaffolds after HT and the sintering processing steps, and also after the BG coating and calcination, was analysed by SEM. From the images displayed in Figure 3b, it can be observed that the internal structure of CB was preserved after HT and sintering.

Moreover, the sintering step promoted the development of grains. The BCP scaffolds were successfully coated with BG, and as confirmed by EDS (Figure 3c), revealed the presence of the doping elements. The amounts of strontium, magnesium and zinc were 1.77, 2.07 and 1.11 mol%. The measured values for Sr, and Mg are comparable to the planned contents (2 mol%), while for Zn only ~55% of the planned amount has been effectively incorporated. Moreover, the coating with the sol-gel derived BG did not obstruct the pores. The surface coating presents some cracks, which can be attributed to the high shrinkage extents that are typically observed for sol-gel derived BG during the dry and calcination steps. The coefficient of thermal expansion of the BG coating material was 2.27 × 10^−6^ °C^−1^, while that measured for the BCP scaffold derived from CB was 8.15 × 10^−6^ °C^−1^ (i.e., approximately 3.6 times higher). However, the cracks observed in the sol-gel coatings are much less attributable to this difference in the coefficients of thermal expansion, but more to the extent of the shrinkage undergone by the coating, especially under the drying step.

### 3.3. Mechanical Properties

The characteristic stress-strain curves are displayed in Figure 4a. It can be seen that the uncoated scaffolds undergo a gradual collapse layer-by-layer, while in the coated ones, this behaviour is less well demarked. The average compressive strengths of the uncoated and coated scaffolds were calculated from the maxima compressive strength values measured during deformation. The results plotted in Figure 4b show that the average compressive strength of BCP scaffolds derived from CB was 0.26 ± 0.02 MPa, while a higher value of 0.36 ± 0.04 MPa was registered for BG-coated scaffolds. This means that the BG coating significantly improves the mechanical performance of the BCP scaffolds. Moreover, it was registered by the Young’s moduli of 0.34 ± 0.04 and 0.53 ± 0.04 MPa for the uncoated and coated scaffolds, respectively.

### 3.4. In-Vitro Bioactivity Test

The material capacity to bond the natural bone upon implantation is normally evaluated by immersion in the SBF solution, which is recognized to be an interesting first approach for predicting the bioactivity of a material [30]. Figure 5 shows the SEM micrographs of the scaffolds after 14 days of immersion in the SBF solution. After this immersion period, the roughness of the coated scaffolds significantly increased in comparison to the uncoated samples. The EDS analysis revealed that the precipitate mainly consists of Ca, P and O. Therefore, the BG coating also improves the in-vitro bioactivity of the scaffolds. These results can be explained considering that the calcium phosphate materials, like hydroxyapatite, are more chemically stable under pH conditions, similar to those existing in the physiological fluid in comparison to BG [3]. This, combined with the ability to form a bone-like HA layer, confer to bioactive glasses their capacity to strongly bond to the living tissues [4,5].

## 4. Discussion

A Sr-, Mg- and Zn-doped sol-gel derived BG was prepared and used to coat BCP scaffolds derived from CB. The amorphous nature of the sol-gel derived BG was demonstrated by XRD (Figure 1) and also confirmed by the FTIR pattern (Figure 2), showing the characteristic Si–O bands. The absence of any calcium carbonate phase in the scaffolds after the HT transformation and the only presence of the XRD patterns characteristic of HA and β-TCP peaks, prove that the original aragonite was for the first time successfully and completely transformed into BCP scaffolds. The transformation of CB was previously studied by Sarin el al. [33], nevertheless a significant amount of CaO was present in the final product when the aim was to preserve the CB internal structure. However, the free CaO tends to react with body fluid to form Ca(OH)_2_, therefore creating a cytotoxic alkaline environment after the implantation in vivo. Moreover, and in agreement with XRD, the FTIR spectrum of BCP scaffold demonstrated the presence of –PO_4_ and –OH groups. Nonetheless, –OH groups are mainly nonexistent due to their absence in the β-TCP phase and the partial de-hydroxylation of HA upon sintering at 1200 °C. The XRD patterns of uncoated and coated scaffolds were identical, indicating that the surface coating remains amorphous and is fine enough to hide the BCP nature of the substrate. With respect to the FTIR spectra, the characteristic Si–O band located between 900 cm^−1^ and 1300 cm^−1^ appeared overlapped with –PO_4_ bands, making it difficult to distinguish between the differences of the uncoated and coated scaffolds.

The µ-CT image of CB (Figure 3a) demonstrated the unique structural features of CB, which are known to be beneficial for bone development [28]. The unique microstructure was preserved after the HT and sintering steps (Figure 3b). Moreover, it is important to highlight that after the BG coating, the interconnected porous structure was not compromised (Figure 3b). Through EDS analysis (Figure 3c), it was possible to observe that the doping elements Sr and Mg were present in the coating layer in amounts that were close to the planned ones. The incorporation extent of Zn was only ~55%. This lower efficiency of zinc for entering in the glass silicate network as the doping element may be related to its less marked network-modifying role in comparison to the alkaline earth elements. It was found recently that Zn^2+^ tends to adopt a mixed tetrahedral and octahedral coordination in the glassy network and copolymerizes with [SiO4] tetrahedra [34]. On average, a Zn^2+^ was coordinated with ~4.9 non-bridging oxygens (NBOs), a lower value in comparison to the alkaline earth elements (~5.9 NBOs for Ca^2+^, and 5.1 NBOs for Mg^2+^). The lower coordination number of Zn^2+^ suggests that it is probably closer to an intermediate role, reflecting its higher field strength with respect to alkaline earth ions.

Apart from this, the BG coating layer exhibited some cracks because of the sol-gel drying and sintering processes. Despite the observed difference between the thermal expansion coefficients of BCP scaffolds (8.15 × 10^−6^ °C^−1^) and BG coating (2.27 × 10^−6^ °C^−1^), the cracks observed in the SEM images of the coated scaffolds cannot be attributed to this difference. In fact, a coating with a lower coefficient of thermal expansion can undergo smaller dimensional changes upon cooling from the heat treatment temperature (700 °C) to room temperature in comparison to those experienced by the substrate. Therefore, these cracks are only due to the considerable shrinkage undergone by the sol-gel coating during the subsequent heat treatment schedules, especially upon the drying step.

Despite the fine thickness of the BG coating, the mechanical properties of the scaffolds improved in a significant way as seen in Figure 4. However, the compressive strength of the coated scaffolds (0.36 ± 0.04 MPa) are within the lower range of the compressive strength of trabecular bone that varies between 0.1 MPa and 16 MPa [35]. This result could be associated with the 92% porosity of the scaffolds. However, this does not seem to be a problem, as the scaffolds are thought of essentially as bone graft fillers. Therefore, they are not subjected to significant load bearing conditions, ven through the mechanical properties could be further improved by applying successive thinner sol-gel derived BG coatings.

The scaffolds exhibited a good in-vitro bioactivity, reflected by the increases in surface roughness after the immersion period of 14 days in the SBF, as can be observed in Figure 5. The less remarkable surface changes that occurred for the uncoated scaffolds can be attributed to their higher chemical stability under pH values, similar to that of physiological fluid [3]. Conversely, bioactive glasses have the ability to release other ionic species that may contribute to oversaturate the SBF solution, inducing the surface dissolution-precipitation reactions [4,5]. Besides these advantages, the various ionic species released from the BG-coating may activate the osteoprogenitor cells and stimulate the biological processes (neovascularisation and angiogenesis) necessary for promoting bone healing [6,7].

More extensive and remarkable surface changes are likely to occur during longer times of immersion in the SBF. The scaffolds are promising for being further in-vitro investigated in cell cultures. These investigations are still underway and will be reported in the near future.

Overall, the surface functionalization of the BCP scaffolds with a sol-gel derived BG enabled the materials to be obtained with higher bioactivity and compressive strength, and with the ability to release ionic species (Sr^2+^, Mg^2+^ and Zn^2+^) that are known to be beneficial for bone development.

## 5. Conclusions

In the present study, porous BCP scaffolds derived from aragonite CB and with the complete absence of any calcium carbonate phase were, for the first time, successfully prepared by HT transformation. This is a novel and very important achievement, since any untransformed calcium carbonate is likely to decompose during the subsequent sintering step, leading to the formation of free CaO. The free CaO, in turn, tends to react with body fluid to form Ca(OH)_2_, therefore creating a cytotoxic alkaline environment after the implantation in vivo. The BCP scaffolds consisting of 63.9% HA and 36.1% β-TCP BCP were coated with a Sr-, Mg- and Zn-doped sol-gel derived BG, which enhanced their bioactivity in the SBF and compressive strength, enabling the accomplishment of the aimed targets. The BG coating was completely amorphous and the incorporation levels of doping alkaline earth ions (Sr^2+^, Mg^2+^) did not differ much from the planned ones, while the incorporation level of Zn^2+^ was noticeably lower (only ~55% relative to the planned one). The unique porous microstructure of raw CB was neither damaged during the HT and sintering steps, nor compromised after the BG coating. The micro-cracked appearance of the sol-gel derived BG coating is solely due to the shrinkages undergone during the drying and sintering processes.

In summary, the porous BCP scaffolds surface functionalized with the sol-gel derived BG coating seem very promising for being further investigated as bone graft materials and constructs for bone tissue engineering.

## Figures and Tables

**Figure 1 materials-12-02711-f001:**
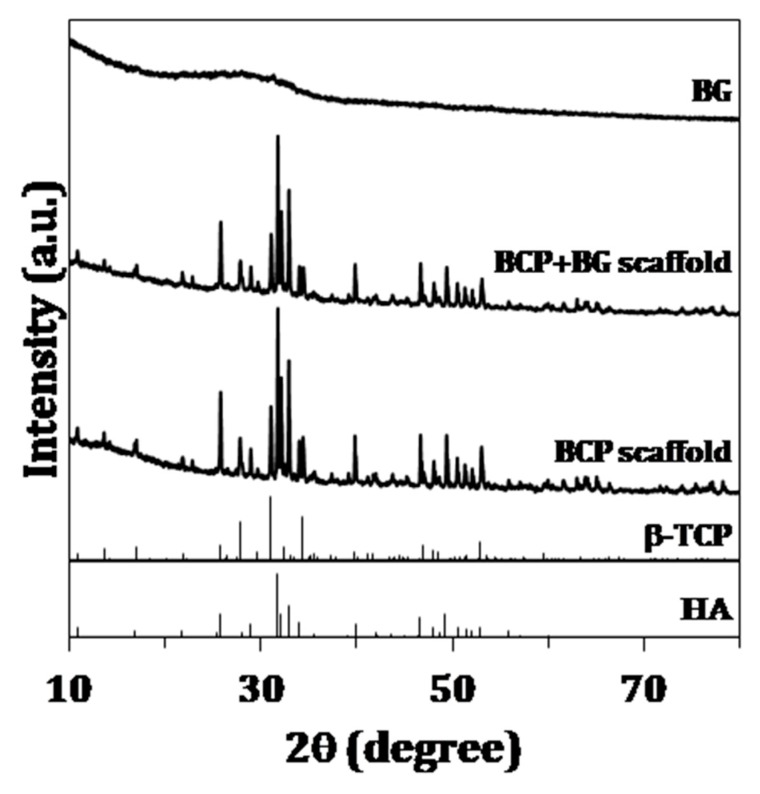
Sol-gel derived bioactive glasses (BG), biphasic calcium phosphates (BCP) scaffolds and BG-coated scaffolds. The standard diffraction patterns of β-tricalcium phosphate (β-TCP) and hydroxyapatite (HA) standards, ICDD PDF 04-006-9376 and 04-015-7245, respectively, are also present for comparison purposes.

**Figure 2 materials-12-02711-f002:**
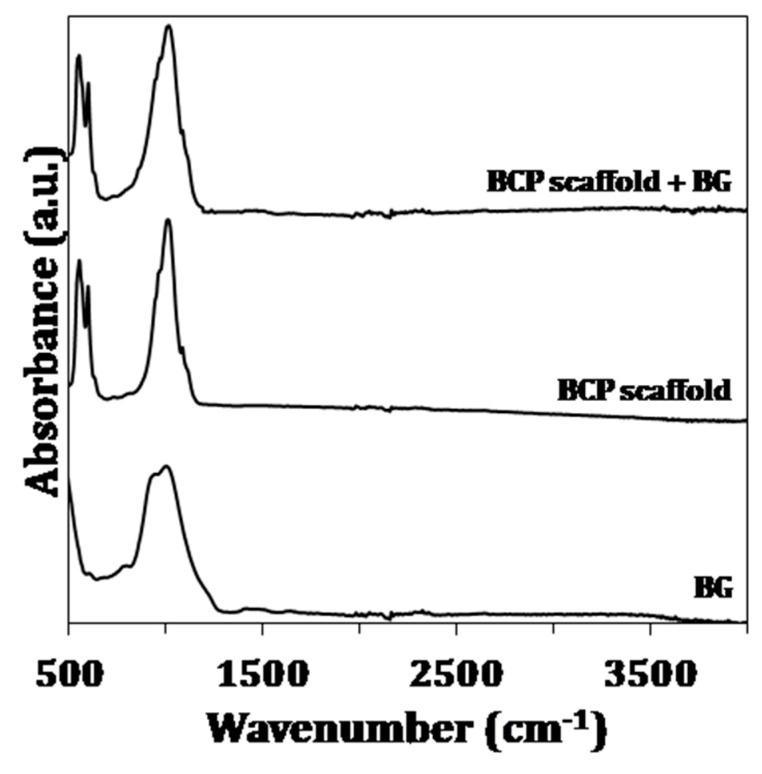
FTIR spectra of sol-gel derived BG, BCP scaffolds and BG-coated scaffolds.

**Figure 3 materials-12-02711-f003:**
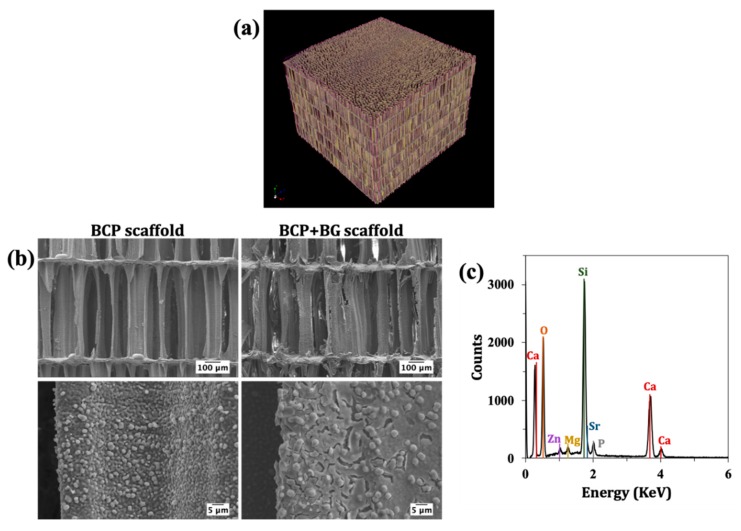
Microstructural and chemical features of the scaffolds: (**a**) µ-CT image showing the microstructure of raw cuttlefish bone (CB) highlighting its interconnected porosity; (**b**) SEM micrographs of uncoated and coated BCP scaffolds; (**c**) energy dispersive spectroscopy (EDS) analysis of the sol-gel derived BG coating.

**Figure 4 materials-12-02711-f004:**
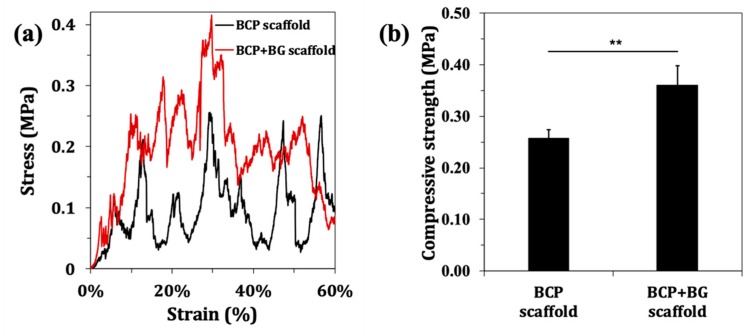
Compressive strength of the uncoated and BG-coated BCP scaffolds: (**a**) representative stress-strain curve; (**b**) maximum compressive strength during the deformation period. The results are represented as the mean ± SD (** P < 0.01).

**Figure 5 materials-12-02711-f005:**
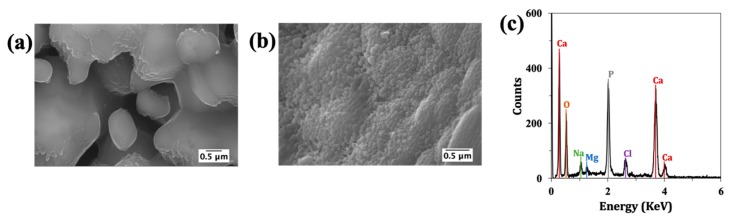
Microstructural and chemical features of the scaffolds after 14 days of immersion in simulated body fluid (SBF) solution: (**a**) SEM micrograph of uncoated BCP scaffolds; (**b**) SEM micrograph of BG-coated BCP; (**c**) EDS analysis of the precipitate.

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
