# Peer review of "Cuttlefish Bone-Derived Biphasic Calcium Phosphate Scaffolds Coated with Sol-Gel Derived Bioactive Glass"

_materials, 2019, doi:10.3390/ma12172711_

Round 1

Reviewer 1 Report

Bioactive glasses as biomaterials have been investigated extensively in bone-tissue engineering, since they offer good biocompatibility and bioactivity. Addition of ions like Si, P and Ca is known to stimulate osseointegration. It is always interesting to see whether different concentration and composition of ions can improve mechanical and biological properties of newly synthetized scaffold. While I have no remarks on material’s chemical and structural characterization and definition of mechanical properties, I believe that the article lacks biological characterization of material.

My remarks are as follows:

1.   In the Material and Methods, in the section 2.4. In vitro mineralization is needed to change capitation into In vitro bioactivity test. It is by no means biomineralization test. The authors have to pay attention through the article and write in vitro in italic.

2.   In the Results, in the section 3.4. In vitro mineralization is needed to change capitation into Bioactivity evaluation. I have feeling that the authors do not differ mineralization from bioactivity. The whole paragraph is overstated, and thus last sentences should be removed. Moreover, it is needed to remove sentences in discussion and conclusion (throughout the whole article) concerning mineralization because mineralization was not evaluated.

Bone mineralization is the process of laying down minerals on a matrix by bone-forming cells. To evaluate mineralization in vitro, the cells are needed (for example hMSCs) which will be responsible for matrix deposition and sequentially mineralization after 14 and 21 days of cell culture.

3.   I suggest at least evaluation of the proliferation capacity of cells (Hek293 or hMSCs or other human healthy cells) by MTT assay during three days. The cells should be seeded on BCP and BCP+BG scaffolds.

4.   It is missing explanation why specific concentrations of ions (i.e. 60% SiO2, 34% CaO, 2% Sro, 2% MgO, 2% ZnO, 2% P2O5) were used.

Author Response

Dear Reviewer,

Our responses point by pint to the comments received from each Reviewer were collected in the single document attached.

Reviewer 2 Report

Comment 1:

The authors employed cuttlefish bone as a scaffold. As the author said, cuttlefish is calcium carbonate and we should transform calcium carbonate to calcium phosphate. Why did the author use calcium carbonate as a precursor, not calcium phosphate? The authors should mention the merit of cuttlefish bone.

Comment 2:

The authors compared biphasic calcium phosphate and bioactive glass coated biphasic calcium phosphate. If they claimed that developed bioactive glass has synergetic effect, they should also compared with bioactive glass alone.

Comment 3:

After immersing SBF, they evaluated surface morphology by SEM and suggest that in vitro bio-mineralization ability was improved. Maybe this is true. However, the authors should identify what the precipitant is.

Comment 4:

There are so many mistype even though this is the submitted manuscript.

Author Response

Our responses point by pint to the comments received from each Reviewer were collected in the single document attached.

Reviewer 3 Report

The comments and suggestions are reported in the attached file.

Author Response

(The authors gave the same response as above.)

Reviewer 4 Report

Despite years long research and development, material with optimal features for bone engineering has not been yet produced. In the present work, Neto and colleagues made a solid step towards achieving this goal by combining hydrothermally transformed natural cuttlefish bone with sol-gel derived bioactive glass. Resulting material was characterized for their physical and mechanical properties, as well as for their morphology and capacity to mineralize human body fluid. Overall, the research contained in this manuscript is well designed, accomplished, described, and commented. The level of innovation (mainly hydrothermal transformation of bone and functionalization by sol-gel derived bioactive glass) is very clear and the procedures as a whole open up new avenues for further optimization.

Author Response

(The authors gave the same response as above.)

Round 2

Reviewer 1 Report

After careful consideration, I suggest to accept the manuscript.

The authors must change the last two sentences in the abstract :

The sentence: “The results of in-vitro bioactivity after immersing the BG coated scaffolds in simulated body fluid (SBF) showed extensive formation of bone-like apatite onto the surface of the scaffolds. “ should be changed into “The results of in-vitro bioactivity after 15 immersing the BG coated scaffolds in simulated body fluid (SBF) showed formation of bioactive surface on the scaffolds.”

The sentence: “Overall, the functionalized CB derived BCP scaffolds revealed promising properties for their use in bone tissue engineering field.” should be changed into “Overall, the functionalized CB derived BCP scaffolds revealed promising properties but further assessment of the in-vitro biological properties is needed.”

Author Response

The authors are thankful to Reviewer 1 for providing useful comments that helped improving the quality and the clarity of our manuscript and for recommending its acceptation for publication.

The last two sentences in the abstract were suitably reworded. The entire manuscript was carefully rechecked.

Reviewer 2 Report

The authors must check the final version of manuscript. For instance, although they write '8.15 x 10-6ºC-1', they should write '8.15 x 10-6ºC-1'. There are many same types of mistake. This may be system error when the author submitted this manuscript to the journal.

Author Response

Thanks for the remark. The entire manuscript was carefully rechecked and the typos corrected.